# From the Skin to Distant Sites: T Cells in Psoriatic Disease

**DOI:** 10.3390/ijms242115707

**Published:** 2023-10-28

**Authors:** Eva Reali, Davide Ferrari

**Affiliations:** 1Department of Translational Medicine, University of Ferrara, 44100 Ferrara, Italy; 2Department of Life Science and Biotechnology, University of Ferrara, 44100 Ferrara, Italy

**Keywords:** memory T cell phenotype, psoriasis, psoriatic arthritis, skin immunology, inflammation

## Abstract

Human skin has long been known as a protective organ, acting as a mechanical barrier towards the external environment. More recent is the acquisition that in addition to this fundamental role, the complex architecture of the skin hosts a variety of immune and non-immune cells playing preeminent roles in immunological processes aimed at blocking infections, tumor progression and migration, and elimination of xenobiotics. On the other hand, dysregulated or excessive immunological response into the skin leads to autoimmune reactions culminating in a variety of skin pathological manifestations. Among them is psoriasis, a multifactorial, immune-mediated disease with a strong genetic basis. Psoriasis affects 2–3% of the population; it is associated with cardiovascular comorbidities, and in up to 30% of the cases, with psoriatic arthritis. The pathogenesis of psoriasis is due to the complex interplay between the genetic background of the patient, environmental factors, and both innate and adaptive responses. Moreover, an autoimmune component and the comprehension of the mechanisms linking chronic skin inflammation with systemic and joint manifestations in psoriatic patients is still a major challenge. The understanding of these mechanisms may offer a valuable chance to find targetable molecules to treat the disease and prevent its progression to severe systemic conditions.

## 1. Introduction

Psoriasis is a chronic–relapsing inflammatory skin disease with a multifactorial etiology, with a strong genetic basis and an autoimmune component. It affects on average 3% of the population and it can appear at any age but is more frequent between the ages of 15 and 30. Although the exact etiology of psoriasis is unknown, according to a widely shared view it can be caused by non-specific triggering factors such as mild trauma, drugs, stress, and bacterial infections which can initiate the inflammatory processes that lead to the development of the disease [1]. Recent studies suggest that a damaging insult to keratinocytes in genetically predisposed individuals can trigger inflammatory pathways that activate the inflammasome NLRP3, resulting in the production of pro-inflammatory cytokines such as IL-1β and CXCL8/IL-8 that mediate the subsequent inflammatory cascade of events leading to the development of clinical manifestations of the pathology [2,3]. The immunological mechanism that drives the amplification of inflammation in the skin is centered on the interaction between dendritic cells (DCs) and T cells that generates a self-sustaining inflammatory cycle around the TNF-α/IL-23/IL-17 axis, underlying the development of psoriatic plaques [4]. Psoriasis is associated with systemic and joint manifestations; in fact, 30% of patients develop psoriatic arthritis (PsA) and comorbidities such as increased risk of cardiovascular diseases and metabolic syndrome [5,6,7,8,9]. For these reasons, it is defined as a systemic disease.

According to the current view, the encounter of one of many potential environmental factors may induce keratinocyte stress leading to the release of self-DNA, self-RNA, and antimicrobial peptide complexes into the extracellular compartment [10]. These complexes activate, via TLR9, TLR7 and TLR8 myeloids (mDCs) that produce TNF-α, IL-23, IL-12, and the plasmacytoid (pDCs) that produce IFN-α. DCs migrate into regional lymph nodes and, by presenting antigens, can activate T cells. T cells play a major role in the pathogenesis of cutaneous psoriasis. Cells with the Th17 phenotype are markedly amplified in psoriatic skin lesions, and IL-17 was also shown as an excellent target for the most cytokine blocking therapies [11,12]. CD8^+^ T cells mainly residing in the epidermis also play a role in the pathogenesis of the disease.

It is, however, unknown the link between T cell responses arising and amplifying in the skin and the development of extra cutaneous manifestations and comorbidities. Of particular interest is the link with psoriatic arthritis, which usually starts from the entheses and in most cases develops 8–10 years after the appearance of the cutaneous symptoms [13,14]. The development of psoriatic arthritis has been shown to be preceded by a long asymptomatic phase in which signs of synovio–entheseal inflammation are evident only by diagnostic imaging techniques [15].

The link between T cells and the asymptomatic phase may provide a new reading key to understanding psoriasis and associated comorbidities, opening new perspectives for therapeutic interventions. Here, we took advantage of the increasing knowledge accumulating in the field of cutaneous immunology to envisage possible physiological mechanisms of skin immunology for the establishment of psoriasis and its systemic manifestations.

We reviewed the most updated literature on T cell mechanisms linking cutaneous and systemic immunity and integrate them with literature data on the role of T cells in psoriatic disease. The scenario that emerges could substantially contribute to the understanding of this complex disease and its manifestations at distant sites.

## 2. T Cells and Immunology of the Skin

The skin is the largest organ of the body and a central barrier to maintain tissue homeostasis and prevent damage upon environmental triggers, as well as microbial assaults. It also represents an exclusive environment in which skin cells interact with immune cells to induce suitable immune responses. Also, the idea that skin cells interact with external microorganisms (bacteria, viruses, fungi, archea) in a delicate equilibrium that needs to be maintained to ensure skin health is gaining momentum. Alteration of entities in skin microbiota–host interactions would thus favor the establishment of potentially pathogenetic conditions [16].

Although skin immune responses are mediated by different immune cell types, including Langerhans cells, mast cells, macrophages, neutrophils, and B and T lymphocytes, the initial trigger is often given by stimulation and activation of skin cells, including keratinocytes, fibroblasts, endothelial cells, and adipocytes. These cellular elements work as sensors by interacting with microorganisms as well as poisonous and irritating molecules, thus alerting the immune effector cells [17,18]. Degenerated and degraded structural cells or matrixes forming molecules can also act as stimuli to initiate a pathological response, ultimately leading to impairment of skin integrity and function.

The type, intensity, and duration of the immune responses largely depend on the nature of the stimuli and type of recruited immune cells. To this aim, the transition of resident structural cells to chemokine-secreting and inflammation receptor-expressing elements is fundamental. Hence, keratinocytes, in addition to synthesizing the major structural and isolating proteins of the skin, i.e., keratin, are endowed with the ability to act as sensor cells, alerting immune cells by releasing alarm signals such as intracellular adenosine triphosphate (ATP). Inflammasome-mediated responses through activation of the ATP receptor P2X7 induces IL-23 secretion [3,19,20]. IL-23, together with IL-12 and IL-17, are among the central cytokines involved in the pathogenesis of psoriasis [21,22]. Aberrant polarization of CD4^+^ T helper lymphocytes induce secretion of TNF-α, IFN-γ, IL-17, and IL-22, activating keratinocytes, thus amplifying inflammation in psoriatic plaques [23].

Cutaneous DCs and keratinocytes are able to perceive tissue damage derived from different sources, through receptors recognizing molecular patterns deriving from pathogens or host-derived molecules exposed following tissue damage [24,25,26,27]. Skin DCs consist of various populations located in the epidermis or dermis and respond to antigens and allergens with the production of chemokines, inflammatory cytokines, and biocidal molecules such as nitric oxide (NO). Langerhans cells, dermal DCs (dDCs), pDCs, macrophages, and the recently identified CD103^+^ DCs represent a fundamental part of the complex puzzle of skin defense, strategically positioned for antigen presentation to circulating but also resident lymphocytes [18,23,28].

CD103^+^ DCs of the type 1-conventional dendritic cell (cDC1) subtype are specialized in cross-presentation of cell-associated foreign antigens and upon activation will migrate to the skin-draining lymph nodes to activate naïve T cells, preparing them to differentiate into effectors [29]. The activated T cells can then migrate to the epidermis to control infection by producing signals that recruit additional immune effector cells. In addition, DCs of the type 2-conventional dendritic cell (cDC2) subgroups play a central role in the induction and maintenance of T cell tolerance of the skin towards both self-antigens and antigens derived from commensal bacteria [30,31].

Healthy skin contains a number of T cells that are more than double than the ones found in peripheral blood [32,33]. Most of them are memory T cells that have previously encountered antigens and can rapidly reactivate. CD8^+^ T cells are mainly found in the epidermis, whereas CD4^+^ T lymphocytes are found mainly in the dermis, where they have the role of amplifying the immune response. Among T cells, γδT cells are known to enhance wound healing by secretion of the insulin-like growth factor 1 (IGF-1) [34,35]. Depending on the Vδ and Vγ gene expressed, these cells contain different subpopulations, showing heterogeneous tissue colonization [36]. Vδ1 γδ T cells are mainly located in the dermis, while Vδ2 T cells are more abundant in peripheral blood and dermis [37]. Inappropriate activation of dermal γδT cells plays a role in psoriasis pathogenesis, since they are endowed with the ability to produce IL-17 [38].

Memory T cells in the blood have been initially classified into central memory that express CCR7 and CD62L high endothelial venule (HEV) homing molecules and constitutively recirculate between the blood and secondary lymphoid organs as well as effector memory T cells found only in the blood and lacking HEV homing molecules [39].

New light has been recently shed on the role of tissue-resident memory T lymphocytes (T_RM_) residing in the skin to grant a prompt response in case of re-infection of the tissue by microorganisms. Both helper CD4^+^ and cytotoxic CD8^+^ T_RM_ have been found in the skin [40,41,42,43,44,45] and also act as responders in autoimmunity as well as in psoriasis and other skin diseases [46]. Resident memory T lymphocytes of the skin may allocate to this organ for years as they stably express CD69 and the integrin alpha E (CD103), which in turn binds E-cadherin expressed by keratinocytes. Moreover, they express skin homing molecules such as cutaneous leukocyte antigen (CLA) and the chemokine receptors CCR4 and CCR10. This is paralleled by down-regulation of membrane molecules involved in cell recirculation between tissues [47]. In the human epidermis there are indeed many CD4^+^ and CD8^+^ T_RM_ [48]. Skin T_RM_ are found at increased level in the upper hair follicles and require IL-7 and IL-15 signaling for their survival [49,50]. In the generation and persistence of T_RM_, a central role is played by transforming growth factor-β (TGF-β) which is required for the CD8^+^ T_RM_ to be retained in the epidermis and mediates its effects through the CD103 induction on T cells [51].

In the epidermis, TGF-β produced by keratinocytes favors the differentiation of skin-recruited CD8^+^ T cells into T_RM_ cells after antigen-specific activation and expansion in the lymph node and it also induces their persistence in the epidermis. Intriguingly, T_RM_ activated by both antigen-specific and bystander activation can persist as T_RM_, however, under limiting conditions, antigen-specific T_RM_ cells were more efficiently retained than bystander T_RM_ cells. This mechanism may represent a selective pressure that favors the persistence and accumulation of antigen-specific T_RM_ cells at the skin barrier and may, on the other hand, set the basis for postulating that a fraction of cells with effectors/terminally differentiated phenotype derived from non-antigen-specific cells may be found in the circulation under chronic inflammatory conditions (Figure 1).

In addition to the prominent role of T_RM_, an indication of a skin-to-blood recirculating subset with T_CM_ phenotype and expressing CCR4 was provided by Luster and co-workers showing, in a Kaede transgenic mouse model, that CD4 memory T cells can egress from the skin in a CCR7-dependent manner. Specifically, the authors characterized a recirculating memory CD4 T cell subset with phenotype CCR7^+^CD69^-^CD103^-^/CCR4^+^ that can enter the circulation and maintain the capability to migrate into normal skin [52]. Circulating CD8 T cells also have the ability to enter the skin in the absence of infection and were identified for their lack of CD69 expression. In support of the physiological role of this mechanism, it has been shown that loss of recirculating memory T cells impairs host defense to skin infection [53]. As an additional characterization, Clark and coworkers in 2015 have defined two distinct populations of CCR7^+^ recirculating T cells: the central memory T cells (T_CM_) expressing L-selectin and CCR7^+^ T cells negative for L-selectin, which were classified as migratory memory T cells (T_MM_) [48]. This circulating skin-tropic T_MM_ subset produced cytokines at an intermediate level between T_CM_ and effector memory T cells.

T_MM_ cells were present in the dermis and absent from the epidermis. A significant fraction of these cells also expressed CD69, suggesting that they may be in a transitional state towards a more differentiated phenotype. About two thirds of T_MM_ in both healthy skin and inflamed tissues, however, lack CD69. These cells, unlike T_CM_ and naïve T cells, lack L-selectin and probably cannot migrate through high endothelial venues to enter the lymph nodes from the blood, but rather migrate directly back to the skin where they could either give rise to T_RM_ or recirculate back to the blood. Possibly, both T_RM_ and recirculating T cells could contribute to the pathogenesis of inflammatory and autoimmune diseases, and understanding all the steps of this process can be critical for the identification of therapies targeting pathogenic T cells involved in specific phases of disease manifestations.

A T cell-intrinsic skin-homing program is established during priming in the context of skin infection and maintained even without antigen re-exposure. According to a recent definition, CCR4 is a skin-homing receptor expressed by memory T cells which have been primed in lymph nodes draining the skin compartment [54,55,56]. Cells expressing CCR4 can enter the skin under normal non-inflammatory conditions, by interacting with low levels of the homing molecules E-selectin and CCL17, constitutively expressed in resting endothelium, that bind CLA and CCR4 on T cells [57,58].

Actually, an earlier study by Campbell and colleagues had already suggested the role of the chemokine receptor CCR4 as a skin-homing molecule mediating the migration of T cells from the blood to the dermis. The study shows that in peripheral blood a considerable number of memory T cells express the chemokine receptor CCR4 and respond to the CCL17 chemokine [59]. The CCL17 chemokine is constitutively expressed by skin epidermal cells and released under inflammatory conditions. In addition, CCR4 also binds to CCL22, which also plays a role in other tissues [60].

Binding of CCR4 to CCL17 causes the arrest of memory T cells under physiological flow integrin-dependent adhesion of skin memory T cells to the cell-adhesion molecule ICAM-1. CCR4/CCL17 axis therefore emerges as a central element for the interaction between circulating memory T cells and the skin vasculature, thus directing memory T cells to their original target tissues [59,61]. The chemokine receptor CCR4 would favor the trafficking of cutaneous memory type 2 Th cells as well as of other T cell phenotypes to the skin [62].

In the CD8 compartment, CCR4^+^ T cells were predominantly found in the CD27^+^CD28^+^CD45RA^−^ memory subset and expressed the CCR7^+^CCR5^−^ phenotype. CCR4^+^CD8^+^ T cells did not express perforin and granzymes A and B that are a feature of differentiated effectors. This evidence has suggested that they were more immature than CCR6^+^CD8 early effector memory T cells that express GraA and perforin and accordingly with more recent study contribute to place these cells in an early stage of the memory T cell differentiation pathway [63].

According to this view, analysis of CCR4 and skin-tropic molecules in the different stages of memory T cell differentiation in CD8^+^, CD4^+^CD45RA^−^CCR7^+^ (T_CM_), and CD45RA^−^CCR7^−^ (T_EM_) cells indicate that the skin-homing CCR4 marker is expressed mainly in cells with a CCR7^+^ phenotype, both L-selectin^+^ and L-selectin^−^, thus including T_CM_ and T_MM_ subsets described by Clark and colleagues [48] CCR4^+^T_CM_/_MM_ cells also express high level of CLA, whereas T_EM_ expresses CXCR3 and CCR5 and lower levels of CCR4 and CLA [64]. This indeed supports the concept that progressive stages of memory T cell differentiation have different chemokine receptor profiles, and that skin-tropic features are progressively lost as the differentiation pathway progresses towards the effector memory/effector phenotype.

In another study, the question of whether CCR4^+^ T_CM_ cells could represent a subset with high plasticity that, upon antigen encounter, can progressively shift their phenotype toward CXCR3^+^ T_EM_ was addressed. Sorted CCR7^+^CD45RA^−^CXCR3^−^ cells from peripheral blood mononuclear cells (PBMCs) stimulated with αCD3/αCD28 beads were analyzed at different time points. On day 3, after T cell receptor (TCR) stimulation, all CCR4^−^-enriched T_CM_ cells shifted their phenotype to double positive CCR4^+^CXCR3^+^ T_CM_, and at later time points a small fraction of cells that were single positive for CXCR3 and a fraction of cells with the T_EM_ phenotype were detected [65].

Under physiological conditions and in early inflammation, CCR4 could therefore be required for lymphocyte trafficking to the skin, whereas under severe inflammatory conditions other chemokines could play a major role in the recruitment of effector memory/effector T cells. Consistent with this view, different groups evidenced that T cell-mediated skin inflammation is largely independent of CCR4 and rather requires CXCR3 [61].

CCR4 central/migratory memory T cells could be crucial in skin patrolling rather than in mediating the recruitment of differentiated effectors cells to the inflamed skin. To explain this phenomenon, a T cell phenotype hierarchy in skin inflammation that places early memory and antigen-specific T cells upstream of the inflammatory cascade and a massive recruitment of CXCR3 effector from the blood as a downstream event of advanced inflammation were postulated [66].

The skin compartment had previously been proposed as a “peripheral lymphoid organ”, important for antigen encounters and lymphocyte differentiation [67]. The evidence accumulated in this field strengthens at least the role of the skin as a preferential trafficking site for T_CM/MM_ in an early stage of differentiation, where it is possible that antigen encounter occurs. These cells after an antigen encounter and exposure to inflammatory stimuli could progress to a non-circulating T_RM_ phenotype [40,68] (model depicted in Figure 1). In immunopathological skin conditions such as psoriasis, the expanded subset of T_CM_ cells expressing CCR4 and CXCR3 could play a role in disease recurrence or redistribution to distant sites such as joint synovial tissues and enthesis.

## 3. T Cells in Psoriasis Pathogenesis

Critical components of the T cell response involved in the initiation and amplification phases of cutaneous psoriasis have been described in the last two decades. It should be mentioned that in the psoriatic skin, CD103^+^ tissue resident memory T cells of CD8 lineage were found by Eidsmo and coworkers, describing the role of CD8 T_RM_ cells with a Tc17 phenotype in specific disease memory in sites of recurrent psoriasis [46,69,70].

As regards to the role of T cells in the early formation of the psoriatic plaque, Bowcock and Krueger proposed a role for the perivascular aggregation of T cells and mature DCs in the dermis which, in this case, could work as an ectopic lymphoid aggregate or, according to a more recent definition, as a tertiary lymphoid structure [71].

Subsequent studies indicated the interaction between the chemokine, CCL19, and its ligand CCR7, which is typical of T cell activation in lymphoid structures as a key event for the recruitment of these cells also in psoriatic skin [72]. In the dermis of psoriatic plaques, the presence of a lymphoid aggregates has been reported by Mitsui and colleagues, evidencing also the expression of *CCL19* and *CCR7* [73]. This first suggested a role for the CCR7/CCL19 axis in disease pathogenesis. At the same time, a study by Bosè and colleagues showed that anti-TNF-α therapy in psoriasis patients led to the inhibition of the CCL19/CCR7 axis in psoriatic skin lesions and that this phenomenon was correlated with disease regression [74]. This importantly enlightened CCR7/CCL19 axis in skin lymphoid aggregates and showed that one of the mechanisms at the basis of the long-term effect of TNF-α-blocking treatment is actually the destruction of these structures. To this end, it is important to mention that among the multiple functions of TNF-α/β they also contribute to the maintenance of the secondary lymphoid organ architecture and promote angiogenesis through the induction of VEGF [75,76,77].

Along this line, data obtained by Canete and colleagues highlighted the presence of lymphoid aggregates in the synovial tissues of patients with psoriatic arthritis and showed that these structures were significantly reduced by anti-TNF-α therapy [78].

Deep phenotyping of the circulating T cell compartment performed in our laboratory showed enhancement in the percentage of memory T cell subsets expressing putative pathogenic double-positive IL-17 and IFN-γ non-classic Th1, described by Annunziato and colleagues [79], and CD8 cells, as compared to healthy subjects. Therefore, it was envisaged that in the dermal lymphoid structures of psoriatic plaques something essential for the pathogenesis of the systemic disease manifestations could happen [80,81,82,83]. In particular in these structure, T_CM_ cells attracted from the blood through the CCR7/CCL19 chemokine axis could be activated either by an antigen encounter or by bystander activation and give rise to both T_RM_ or T_CM/TMM_ that egress the skin with an inflammatory phenotype, but at different stages of memory differentiation.

Then, a series of studies were developed to investigate the hypothesis that skin-to-blood recirculation of T cells could take part in the development of psoriatic arthritis and comorbidities [84].

## 4. T Cells in the Pathogenesis of Psoriatic Arthritis

Psoriatic arthritis develops in up to 30% of patients with cutaneous psoriasis and in the majority of cases it follows the development of the cutaneous manifestations for 8 to 10 years [85].

Enthesitis is indeed a feature of psoriatic arthritis and it is increasingly evident that, in the joints of patients with psoriatic arthritis, inflammation can start from the entheses, the attachment sites of ligament to bone, during a subclinical phase of the disease [86,87]. In the subclinical phase of entheseal, inflammation psoriatic arthritis (PsA) is silent and is only evidenced by diagnostic imaging techniques such as ultrasonography, magnetic resonance imaging (MRI), and computed tomography that have empowered the concept of an evolution from cutaneous to synovio-entheseal inflammation [13,14,15]. This evidence also opened the possibility to investigate the events linked to the development of psoriatic arthritis in patients with psoriasis.

For a better understanding of the link between cutaneous psoriasis and psoriatic arthritis, it is important to underline that psoriasis has an autoimmune component and T cells reactive to autoantigens cathelicidin LL-37, melanocytic ADAMTSL5, lipid antigen PLA2G4D, and keratin 17 have been identified in patients’ peripheral blood [88,89,90]. The possibility that activated T cells reacting or cross-reacting with a self-component may enter the bloodstream and reach distant organs should therefore be considered as a causal link with joint manifestations and systemic inflammation.

In patients with psoriatic arthritis, Curran and colleagues analyzed the clonotype of T cells in the inflamed joint, peripheral blood, and skin samples by sequencing theTCRβ chain genes [91]. This pivotal work identified three populations of T cells in joint tissues: one highly represented population of polyclonal CD4 T cells that did not persist in the tissue after treatment with methotrexate, a second population of moderately expanded inflammation-related clones either of CD4 or CD8 lineage, and finally, a small population of highly expanded clones which were only of the CD8 lineage. These CD8 T clones showed a marked expansion in both peripheral blood and synovial fluid and persisted during methotrexate treatment.

These expanded CD8 T cell clones were proposed as potential drivers and could link psoriasis and PsA [92,93,94].

As regards the major polyclonal CD4 T cell population that has been described both in blood and in inflamed tissue, it could potentially represent a recirculating population of the Th17/Th1 subset originating in psoriatic plaques that traffic to the joints [91]. On this basis, the previously mentioned hierarchy in the T cell-mediated cascade of events characterizing psoriatic disease, comprising an autoimmune clonally expanded CD8^+^ T cell subset followed by Th17 and Th1 CD4^+^ T cells expanded either by cross-stimulation or by bystander activation and a major downstream recruitment of CXCR3^+^ T cells with different specificities induced by the increased expression of the chemokine CXCL10 has been hypothesized [95]. Accordingly, our group and a parallel study of scRNAseq showed accumulation of T cells with CXCR3 and an effector phenotype in the synovial fluid, paired by a massive increase in the CXCL10 chemokine which binds CXCR3 [81,96].

Given the amount of evidence collected in the field of cutaneous immunity in physiological conditions, the concept could be extended to skin immunopathology, by hypothesizing that in human diseases of barrier tissues, such as in psoriasis, a similar dynamic balance between tissue resident memory T cells and the pool of recirculating T cells could be involved in the pathogenesis of the disease [75].

Consistent with the role enlightened for CCR4 in skin-homing T cells and in the cells egressing the skin with a T_CM_ phenotype, CCR4^+^ CD4 T cells in the circulation correlated positively with the severity of the cutaneous disease [82,97]. CCR4^+^ memory T cells could therefore represent a recirculating population responsible for systemic manifestations associated with severe psoriasis.

By contrast, analysis of the possible link between skin psoriasis and systemic inflammation indicates mainly a correlation with the percentage of CD8 T cells in a terminally differentiated phenotype (T_EM_ and T_EFF_). This has been observed in different studies, evidencing a correlation between CCR4^+^ CCR5^+^ CD8 T_EFF_ with serum level of C-reactive protein (CRP) and with (psoriasis area and severity index) PASI score in patients with psoriatic disease. Accordingly, in patients with psoriasis and PsA, a correlation between CD8 T_EMRA_-expressing CD69 early activation markers and the level of serum CRP was observed. Regarding CD4 T cells, a positive correlation with serum CRP was observed only with the percentage of CCR6^+^ terminally differentiated effector memory cells re-expressing CD45RA (EMRA) in patients with psoriasis. This suggests an involvement of terminally differentiated CD8 T cells, and to a minor extent of Th17 T_EMRA_, in the systemic inflammatory state [82,97]. In patients with PsA, analysis of blood and synovial fluid memory CD4^+^ and CD8^+^ T cells indicates a shift from the CCR4^+^ phenotype to CCR4^+^CXCR3^+^ and CXCR3^+^CCR4^-^, respectively [81].

Particularly intriguing is the link between psoriasis and PsA.

In light of the role that emerged for CCR4 in the recirculation of memory T cells forming the skin to the peripheral blood, we addressed the possibility that CCR4 cells are clonally linked to the self-reactive clones found in the blood and joint tissues of PsA patients.

To understand the mechanistic and clonal link between skin-primed T cells and clone expansion in the peripheral blood of psoriatic disease patients or in the synovial fluid of PsA individuals, an imiquimod-induced psoriasis-like inflammation model in K5-mOVA.tg mice expressing ovalbumin on the membrane of keratinocytes adoptively transferred with ovalbumin-specific OT-I.tg naïve CD8^+^ T cells was used to add the antigen-specific T cell component to the model of psoriasis-like inflammation [98].

In this mouse model, it was shown that prolonged skin inflammation induced by imiquimod favors the activation of OT-I CD8^+^ T cells specific to the cutaneous self-antigen [65]. These cells had a CCR4^+^ phenotype and were found in peripheral blood and in an expanded pool of memory T cells in the spleen. Therefore, the mechanistic evidence is that psoriasis-like skin inflammation induced by imiquimod-activated T cells specific to cutaneous self-antigens recirculate from skin to blood to spleen.

To understand if an analogous mechanism could play a role in spreading tissue damage and inflammation to the joints in human PsA, we analyzed scRNA-seq data paired with TCR αβ chain sequencing in paired samples of PsA patients’ peripheral blood and synovial fluid CD8 T cells [96]. Clonotype analysis showed a clonal expansions in the CCR4^+^ T_CM_ subset in the peripheral blood of PsA patients as compared to their CCR4 negative counterpart; moreover, there was a clonal link between these expanded CCR4^+^ clones in the blood and clones found in the expanded CXCR3^+^ effectors in the synovial fluid of PsA patients [65]. This finding reinforces the importance of antigen-specific cells (likely self-antigen-specific T cells) entering the systemic circulation in generating inflammation and damage to the joints of PsA patients, with a possible shift in their phenotype towards more differentiated and cytotoxic effectors (Figure 2).

Together, this evidence sheds light on the possibility to design alternative therapeutic strategies, aimed at both preventing the exit of activate/antigen-specific T cells from the skin and/or their re-localization to a distant site. This could be achieved either by intervening in the early phase of T cell activation, thus preventing amplification or by inhibiting re-localization. This latter approach may represent a challenge, as one should be able to specifically inhibit specific recirculating/potentially autoreactive subsets without altering the physiological immune responses.

## 5. Conclusions

In this review, we collected results from studies providing evidence that T cells primed in the skin-draining lymph nodes and expanded during the formation of psoriatic plaques may form a tissue resident memory T cell population specific to cutaneous self-antigens or cross-reactive microbial antigens or may enter the circulation with a CCR7^+^ T_CM_ phenotype and spread cross-reactive and activated T cells through peripheral blood to a distant site. The clonotypic link between T cells in peripheral blood expressing skin-homing chemokine receptor CCR4 and the effector cells expanded in PsA synovial fluid supports the concept that T cell egressing from the skin and trafficking to the joints can play a role in the evolution from cutaneous to synovio-entheseal inflammation.

## Figures and Tables

**Figure 1 ijms-24-15707-f001:**
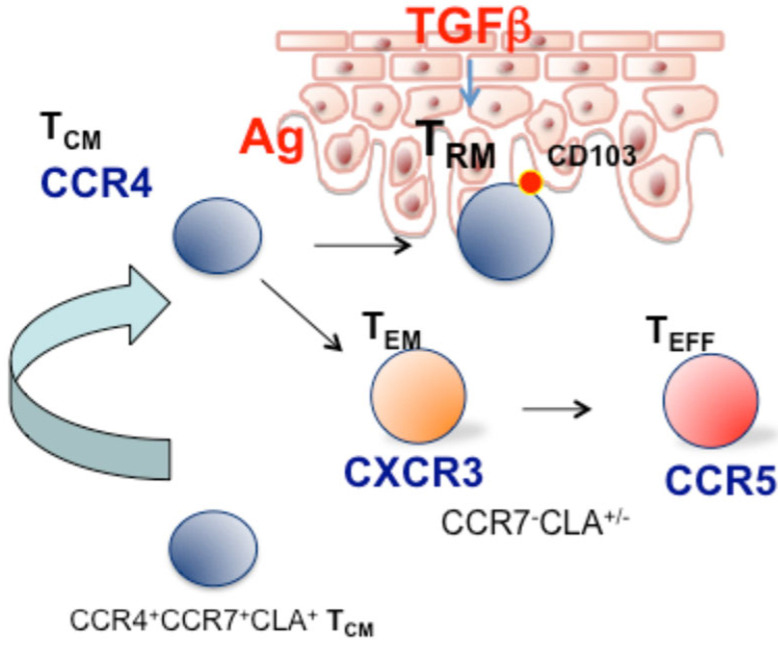
Role of circulating CCR4^+^ CCR7^+^ T_CM_/T_MM_ cells in skin patrolling. After antigens encounter or are under appropriate inflammatory conditions, (i.e., TGF-β), CCR4^+^ CCR7^+^ T cells recruited peripheral blood acquire a long-lived T_RM_ phenotype. In the long term, antigen-specific T_RM_ may accumulate, whereas T_RM_ activated by bystander activation may lose the resident phenotype and be found in the circulation with CXCR3^+^, CCR5^+^ effector/EMRA phenotype.

**Figure 2 ijms-24-15707-f002:**
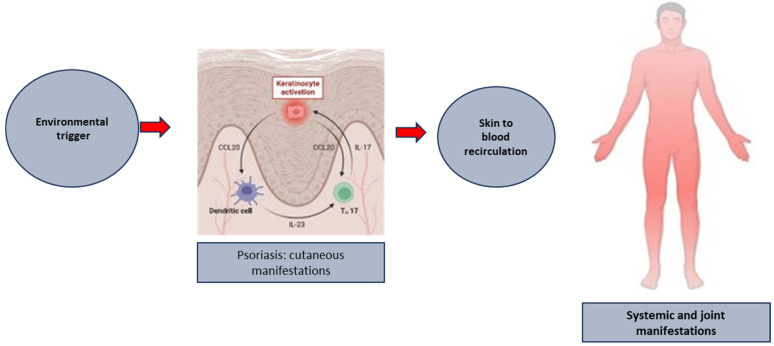
The putative link between the formation of established psoriatic plaques and the development of systemic and joint manifestation of the disease.

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
