# Peer review of "From the Skin to Distant Sites: T Cells in Psoriatic Disease"

_ijms, 2023, doi:10.3390/ijms242115707_

Round 1

Reviewer 1 Report

Comments and Suggestions for Authors It would be important to correctly mark the abbreviations (when they first appear in the text). This would make the text easier to understand even for those who do not read dermatological articles every day.

For example:

“PBMCs” line 176

“TCR stimulation” line 177

“PsA” line 252

„EMRA” line 306

It would also be better if several, complex sentences were written into more simple sentences.
For example:

„The chemokine receptor CCR4 with the trafficking of cutaneous memory type 2 Th cells to the skin [64] [65] however later studies by different groups however do not point towards a role for CCR4 in the trafficking of T cells with a specific polarization phenotype but rather confirmed a role in mediating the homing of the most frequent phenotypes found in the skin [66-68].” lines 151-155

 „Together these data indicate the importance of terminally differentiated CD8 T cells and to a minor extent of putative Th17 terminally differentiated CD4 TEMRA implicated in the pathogenesis and the systemic inflammatory state (C-Reactive Protein) [54,89,110,111].” lines 306-309

Some expressions and abbreviations are not uniform. For example:

“dermis” and “derma” lines 71 and 76

“TCM” and “TCM” lines 177 and 178

There are also some typos in the text.

For example:

“would healing”- ? “wound healing” line 73

- missing period at the end of the sentence: line 73, line 297

“through the induction CD103 on T cells” line 96,97

“the data by obtained”- ? “the data obtained by” line 230

“reported analyzed the clonotype of T cells”- ? “reported the clonotype of analyzed T cells” line 265

“clones expansion”- ???, line 321

“This was mechanistic evidence that...”- ???, line 329,330

“clonotype analysis showed a clonal expansions” , line 336,337

Finally, I have a question for the authors of the article (and I congratulate them on their comprehensive work):

What therapeutic conclusions (if any) can be drawn from the information summarized in the article?

Comments on the Quality of English Language

It would be nice if a native speaker could read the article. It is also understandable in this form, but there are some spelling errors.

Author Response

We are very thankful to the reviewer for the thorough revision of the MS and for useful advices. We have now amended the mistakes present in the submitted MS and revised it accordinghly to reviewer's suggestions.

Davide Ferrari and Eva Reali

Reviewer 2 Report

Comments and Suggestions for Authors

The paper addresses several challenging viewpoints related to the outline of T cells in psoriasis and psoriatic arthritis, as important cellular players and contributors to the mechanisms linking chronic skin inflammation to systemic manifestations in these severe systemic conditions.

However, I would recommend reorganizing the structure of the manuscript, for example inserting a short Introduction section. I would also suggest dividing the first section into two subsections, e.g., related to skin immunology (i) and T cells in skin (ii).

The same suggestion for section 2, where some subdivisions for T cells role in psoriasis (i) namely, in psoriatic arthritis (ii) would be of great help for the reader.

In addition, I suggest to be inserted by the authors a subsection before the Conclusion, in which they should elaborate a little on the therapeutic view of T cells in psoriasis and psoriatic disease.

Author Response

We are grateful to the reviewer for useful advices. We have now submitted a revised version of the MS.

Best Regards,

Davide Ferrari and Eva Reali

Round 2

Reviewer 2 Report

Comments and Suggestions for Authors

The authors have addressed all the issues raised by the reviewer. The manuscript has been improved and modified accordingly. Some references seem a bit outdated, but they are probably meaningful for the subject addressed in the article (e.g., Costello P, 1999).